

**Geoscientific**
**Model Development**

# The effect of accounting for public holidays on the skills of the atmospheric composition model SILAM v.5.7

**Yalda Fatahi**[1], **Rostislav Kouznetsov**[1,2], and **Mikhail Sofiev**[1,2]

[1]Finnish Meteorological Institute, Helsinki, Finland
[2]A.M. Obukhov Institute CE1 for Atmospheric Physics, Moscow, Russia

**Correspondence:** Yalda Fatahi (yalda.fatahi@fmi.fi)

**Abstract.** This study quantifies the impact of emission changes during public holidays on air quality (AQ) and analyses the added value of accounting for the holidays in AQ modelling. Spatial and temporal distributions of atmospheric concentrations of the major air pollutants (the main focus was on $NO_2$, but we also included $O_3$, CO, $PM_{2.5}$, and $SO_2$) were considered at the European scale for all public holidays of 2018. Particular attention was paid to the events with the most pronounced continental- or regional-scale impact: Christmas and New Year, Easter, May Day vacations, and the last days of Ramadan. The simulations were performed with the chemistry transport model SILAM v.5.7 (System for Integrated modeLling of Atmospheric coMposition). Three model runs were made: the baseline with no treatment of holidays, the run considering holidays as Sundays, and the run forcing 80 % reduction in emissions during holidays for the weekday-sensitive sectors. The emission scaling was applied on a country basis. The model predictions were compared with in situ observations collected by the European Environment Agency. The experiment showed that even conservative treatment of official holidays has a large positive impact on $NO_x$ (up to 30 % of reduction in the bias inhomogeneity during the holiday days) and improves the CO, $PM_{2.5}$, and $O_3$ predictions. In many cases, the sensitivity simulations suggested a greater emission reduction than the level of Sundays. An individual consideration of the holiday events in different countries may further improve their representation in the models: specific diurnal pattern of emissions, additional emission due to fireworks, and different driving patterns.

## 1 Introduction

Air quality (AQ) and its temporal and spatial changes are determined by human activities via the release of various air pollutants (Derwent and Hjellbrekke, 2012; Fu et al., 2020; Hassan et al., 2013; Karl et al., 2019; Kukkonen et al., 2020; Lehtomäki et al., 2018; Shi et al., 2019) and modulated by meteorological conditions (Jacob and Winner, 2009; Jhun et al., 2015; Singh et al., 2013; Sofiev et al., 2020).

The ability of atmospheric composition models to follow the temporal variability of air pollution critically depends on the representation of temporal emission profiles by inventories used by the models. Arguably the most difficult task in this context is to reproduce the variations originating from rare irregular events. Changes in human behaviour during non-working days of various type (Beirle et al., 2003; de Foy et al., 2020, 2016; Elansky, 2020; Gour et al., 2013; Hassan et al., 2013; Xu et al., 2017; Zou et al., 2019; Rozbicka and Rozbicki, 2016), including some religious ceremonies (Dasari et al., 2020), cultural practices (Khezri et al., 2015; Nodehi et al., 2018; Ye et al., 2016), celebratory events, and festivities (Hoyos et al., 2020; Jiang et al., 2015; Lai and Brimblecombe, 2017; Retama et al., 2019), cause large variations in emissions of air pollutants, which are hard to quantify and generalize. However, the weekend and (some) holiday effects have certain similarities, which might allow drawing an analogy between weekday vs. weekend and holiday vs. non-holiday pollution levels.

The majority of currently available emission inventories are built as gridded yearly or monthly totals for the key primary pollutants (Frost et al., 2013; Granier et al., 2019, 2011), (https://eccad.aeris-data.fr/, last access: 20 October

**Published by Copernicus Publications on behalf of the European Geosciences Union.**

2021). Temporal variations at shorter timescales have received less attention, but their impact on AQ itself and the model's ability to reproduce the observed concentrations have been considered in several studies (Fu et al., 2013; Gioli et al., 2015; Guevara et al., 2017, 2021; Iriti et al., 2020; McGraw et al., 2010). In particular a crucial role of spatial and temporal resolution of emission inventories for the model's skill scores has been demonstrated (Frost et al., 2013; Gioli et al., 2015; Zhao et al., 2015; Zhou et al., 2020).

Many observation-based studies have been focused on the effects of weekends and, sometimes, specific holidays on pollutant concentrations (Chen et al., 2019; Forster and Solomon, 2003). Lonati et al. (2006) examined the weekend effect on particulate matter ($PM_{10}$ and $PM_{2.5}$) emissions from traffic sources in the city of Milan. The research indicated that concentrations of these compounds in the urban area were lower than the levels on weekdays. Gour et al. (2013) considered differences in pollution levels during weekends and on weekdays in Delhi and showed that the patterns follow the working activities of weekends and weekdays. Parra and Franco (2016), pointed out that the concentration of $NO_2$, $NO_x$ TS1, CO, and $PM_{2.5}$ on working days is higher than that at weekends but the concentration of $O_3$ on working days is lower than that of the weekend, due to ozone titration. In 2017, Ding et al. (2017) CE2 reported that during the Chinese New Year the $NO_x$ emissions are usually lower by about 10 % reflecting the lower business and industrial activities. In a recent study, Hua et al. (2021) estimated the holiday effect on $PM_{2.5}$ and $NO_2$ levels in Beijing by a generalized additive model at 34 air quality monitoring stations during the five heating seasons from 2014 to 2019. According to their results, the holiday effect was much stronger than the weekend effects with increasing $PM_{2.5}$ by 2 % to 30 % but decreasing $NO_2$ concentrations.

Khalil et al. (2016) analysed hourly measurements of $NO_x$, non-methane hydrocarbons (NMHCs), ozone ($O_3$), sulfur dioxide ($SO_2$), $PM_{2.5}$, and $PM_{10}$ collected in the coastal town of Yanbu, Saudi Arabia, during weekends, Eid, Ramadan, and the Hajj periods and demonstrated that the ozone concentrations remained practically the same over these holidays despite the precursor levels being significantly lower. They reported a substantial increase in night-time emissions during Ramadan due to the shift in human activities to night-time.

The fireworks and bonfires during Christmas and New Year of 2013 and 2014 were recognized as the main sources of $PM_{2.5}$ in Mexico city by Retama et al. (2019). Singh et al. (2019) also considered the impact of fireworks on air quality, visibility, and human health and reported significant changes in the pollutant concentrations and a decrease in visibility. Yao et al. (2019) studied air quality trends and firework impact in Shanghai during spring festivals from 2013 to 2017. A decreasing trend in $PM_{2.5}$ in this study revealed the positive effect of the firework regulation on air quality.

Recently, various methods based on observed data and models have been applied to measure the impact of the COVID-19 lockdown on air pollution. These studies investigated the role of transport and industry sectors on pollutant concentrations during the lockdown (Fan et al., 2021 TS2; Grivas et al., 2020; Huang et al., 2020; Menut et al., 2020; Sharma et al., 2020; Wang and Su, 2020).

The above works showed that the effects of isolated events, such as public holidays, can be substantial. Yet their CE3 analysis at large scales (e.g. a continent and a full year) is missing, and a systematic approach to their incorporation into AQ models is yet to be developed.

The goal of the current paper is to address this gap and to take the first step towards incorporation of the public holidays into the regular atmospheric composition and air quality modelling in Europe. We quantified the added value of a comparatively primitive and conservative way of including official holidays into temporal profiles of the emission of air pollutants. Secondly, a sensitivity study was performed demonstrating the extent of the necessary adjustments and potential benefits of a more detailed region-specific analysis of each specific holiday event.

The paper is organized as follows. The next section presents the methodology of the study: information on the European holidays, ways of their incorporation into the emission temporal profiles, the atmospheric composition model SILAM v.5.7 (System for Integrated modeLling of Atmospheric coMposition), and its setup, as well as the statistical measures quantifying the holiday effect. The Results section presents the outcome of the annual SILAM computations for 2018 and the impact of the holiday information on the model skills. The Discussion section compares the outcome with other studies and demonstrates the sensitivity of the results to the changes in the holiday emission representation.

## 2 Materials and methods

### 2.1 European holidays

We collected a list of official holidays in Europe from the Calendarific global holidays API (https://calendarific.com/api-documentation?v=2, last access: 20 October 2021) for the full year of 2018. We regarded the events marked with "national holiday", "local holiday", or "common local holiday" as holidays (see examples for some European countries in Tables 1–3). Since the Sunday emission scaling was applied country-wise, the "local" or "common local" holidays might sometimes cover wider territories than they should. However, it was not possible to accommodate higher level of detail technically, and the choice was between missing some local/regional CE4 holidays and covering wider areas than needed for some events. Since "religious" and "observance" holidays were not considered, we preferred to include the others. The maximum possible error does not ex-

**Table 1.** Official holidays for the example of Finland, 2018. TS3

| Date | Name of holiday |
|------|-----------------|
| 1 January | New Years' Day |
| 6 January | Epiphany |
| 30 March | Good Friday |
| 2 April | Easter Monday |
| 1 May | May Day |
| 10 May | Ascension Day |
| 22 June | Midsummer Eve |
| 23 June | Midsummer |
| 3 November | All Saints' Day |
| 6 December | Independence Day |
| 24 December | Christmas Eve |
| 25 December | Christmas Day |
| 26 December | Boxing Day |

**Table 2.** Official holidays for the example of Germany, 2018.

| Date | Name of holiday |
|------|-----------------|
| 1 January | New Years' Day |
| 30 March | Good Friday |
| 2 April | Easter Monday |
| 1 May | May Day |
| 10 May | Ascension Day |
| 21 May | Whit Monday |
| 3 October | Day of German Unity |
| 25 December | Christmas Day |
| 26 December | Boxing Day |

ceed 10 % because in 2018 national holidays accounted for ∼ 800 country days, whereas common local and local were ∼ 60 and ∼ 80 country days, respectively.

The model computations included all holidays in 2018 but, for the sake of brevity, the analysis below will concentrate on the Christmas and New Year weeks, Easter, and May Day (analysed at the European scale) and the Festival of Breaking the Feast at the last days of Ramadan (Eid al-Fitr, analysed for Turkey).

## 2.2 Atmospheric composition model SILAM

SILAM (System for Integrated modeLling of Atmospheric coMposition, http://silam.fmi.fi/, last access: 20 October 2021) is an offline 3D chemical transport model (Sofiev et al., 2015a), also used for emergency decision support (Sofiev et al., 2006) and inverse atmospheric composition problems (Sofiev, 2019; Vira and Sofiev, 2012). The model incorporates Eulerian and Lagrangian dispersion frameworks and a variety of chemical/physical transformation modules covering the troposphere and the stratosphere (Carslaw et al., 1995; Damski et al., 2007; Gery et al., 1989; Kouznetsov and Sofiev, 2012; Sofiev, 2002, 2000; Sofiev et al., 2010; Yarwood et al., 2005). SILAM features a mass-conservative

**Table 3.** Official holidays for the example of Turkey, 2018.

| Date | Name of holiday |
|------|-----------------|
| 1 January | New Year's Day |
| 23 April | National Sovereignty and Children's Day |
| 1 May | Labour and Solidarity Day |
| 19 May | Commemoration of Atatürk, Youth and Sports Day |
| 15 June | Ramadan Feast |
| 16 June | Ramadan Feast Day 2 |
| 17 June | Ramadan Feast Day 3 |
| 15 July | Democracy and National Unity Day |
| 21 August | Sacrifice Feast |
| 22 August | Sacrifice Feast Day 2 |
| 23 August | Sacrifice Feast Day 3 |
| 24 August | Sacrifice Feast Day 4 |
| 30 August | Victory Day |
| 29 October | Republic Day |

positive–definite advection scheme based on principles laid down by Galperin et al. (1996). The model can be run with various resolutions and coverages starting from a kilometre scale over a limited area and up to the whole globe (Brasseur et al., 2019; Kouznetsov et al., 2020; Petersen et al., 2019; Sofiev et al., 2020, 2015b; Xian et al., 2019). The vertical structure of the modelling domain consists of stacked layers starting from the surface. The layers can be defined either in $z$ or hybrid sigma–pressure coordinates. The model can be driven with a variety of numerical weather prediction or climate models.

## 2.3 Simulation setup

The simulations were performed for the whole year of 2018 for the European domain with the setup following the operational configuration of SILAM in the Copernicus Atmospheric Monitoring Service (CAMS) regional air quality forecasts, as of November 2020 (https://atmosphere. copernicus.eu, last access: 20 October 2021). The only exception was a twice coarser grid resolution to reduce the computational costs (Table 4).

The anthropogenic emissions in the CAMS_REG_AP v4.2 inventory were used as maps of annual totals separately for each country and 16 GNFR sectors (Gridded Nomenclature For Reporting, European Environment Agency, 2013). To obtain the hourly emissions, the annual means were scaled with three temporal profiles, defined separately for each sector, corresponding to month of year (MOY), day of week (DOW), and hour of day (HOD) (Granier et al., 2019). In the CAMS-regional operational setup, the anthropogenic emissions are used without accounting for public holidays.

To assess the sensitivity of pollutant concentrations during holidays, three SILAM runs were made: the baseline with no special holiday treatment (hereinafter, the BL case) and with the holiday days regarded as Sundays (the HS case) and a sensitivity test run with 80 % of emission reduction

**Table 4.** SILAM setup.

| TS5 | |
|---|---|
| Domain and resolution | 30–72° N, 25° W–45° E TS6, 350 × 210 cells of 0.2° × 0.2° size |
| Vertical structure | Ten stacked layers with upper boundaries at 25, 75, 175, 375, 775, 1500, 2700, 4700, 6700, and 8700 m above surface |
| Boundary conditions | First-day operational C-IFS (Integrated Forecasting System of European Centre for Medium-Range Weather Forecasting, ECMWF, with online-coupled chemistry) forecasts at 0.4° resolution |
| Meteorological driver | First-day operational IFS forecasts interpolated to 0.2° × 0.2° regular long–lat grid |
| Anthropogenic emissions | CAMS_REG_AP v4.2/2017 with GNFR temporal and vertical profiles (https://eccad.aeris-data.fr/, last access: 20 October 2021) |
| Natural emissions | SILAM sea salt (Sofiev et al., 2011), dynamic biogenic emissions based upon Poupkou et al. (2010), mineral dust |
| Chemical and aerosol transformations | Modified CBM-5 gas-phase transformation, $SO_4$, $NO_3$, $NH_4$ ion chemistry, $SO_2$ oxidation, nitrate formation, volatility basis set for secondary organics |
| Deposition | Dry: resistance approach (Wesely, 1989) for gases, (Kouznetsov and Sofiev, 2012) for aerosols Wet: SILAM v2018 wet-deposition scheme |

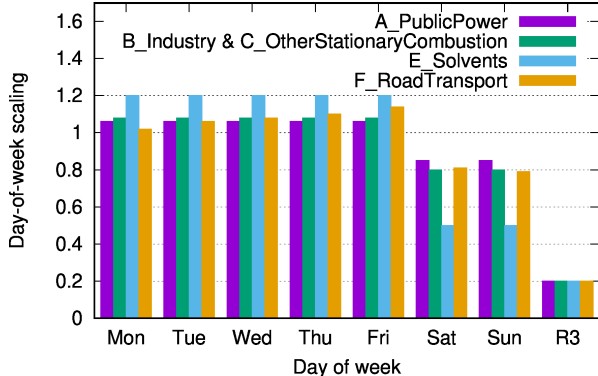

**Figure 1.** Day-of-week coefficients for the affected sectors. R3 is the value forced for national holidays for the R3 case.

during holidays (the R3 case). The emission scaling for the HS and R3 cases were applied only to the sectors affected by the DOW profile. The R3 case was constructed for the Discussion section as a definite low boundary of the possible holiday effect with no realistic scenario behind it. Technically, the emissions were adjusted by altering the DOW scaling coefficients for dates and countries where the holidays occur. For the HS case the coefficients were set to their Sunday values, and for the R3 case they were forced to 0.2. The DOW coefficients for the affected sectors are shown in Fig. 1. Other sectors (D_Fugitives, G_Shipping, H_Aviation, I_OffRoad, J_Waste, K_AgriculturalLivestock, and L_AgriculturalOther) have unity DOW coefficients for all three cases.

## 3   Evaluation scores

For an evaluation of the simulations, we used the hourly data of the AQ monitoring stations downloaded from the European Environmental Agency portal (EEA, http://discomap.eea.europa.eu/map/fme/AirQualityExport.htm, last access: 20 October 2021). Since we focus on regional-scale effects, a subset of representative stations was selected, namely, the stations classified from 1 to 7 according to the Joly and Peuch (2012) classification. This dataset is also used for the operational CAMS-regional evaluation (751 stations over the European domain). For the Ramadan analysis, only Turkish stations were used, with no classification-related filtering applied to maintain a sufficient number of stations in the analysis.

The effect of holidays was considered for the main pollutants observed by the EEA network: $PM_{2.5}$, $SO_2$, CO, $NO_2$, $NO_x$, and $O_3$. Five statistics were considered following the CAMS evaluation standards: bias, fractional bias (FracB), Pearson correlation coefficient (corr), RMSE, and fractional gross error (FGerr).

We considered the effect of holidays at two temporal scales. The short-term impact was analysed for the 1–2-week-long period centred around each holiday day. For each day of this period, the spatial statistics were computed across the observational stations, and the evolution of these statistics from day to day was compared between the SILAM runs. The long-term longitudinal effect was analysed at an annual level for the whole of 2018, and attention was given to the temporal statistics computed for the stations time series.

Since the diurnal profile of emission during the holidays is unknown and probably specific to each event and country, the current study mainly used daily averaging of both observational and model data for computations of the statistics.

Assessing the effect of holidays on the model skills is not straightforward because the emission error during holidays (e.g. too high $NO_x$ emissions) can offset the general under-estimation of the emission in the region as well as the model internal uncertainties. As a result, the model results without the holiday effect may be even better than with it – but for wrong reason. To avoid this problem, we considered the variability of the time series of the model skills as the main measure of success. For instance, a correctly represented holiday effect would lead to the same model bias during the holiday day as before and after. A quantitative measure of success is therefore the ratio $R$ of the standard deviations of the HS and BL runs:

$$R\_P = \frac{\text{SD}^{\text{TS7}}(P_{\text{HS}})}{\text{SD}(P_{\text{BL}})}, \qquad (1)$$

where $P$ is one of the above CAMS spatial model skills and standard deviation is taken from among the daily values of this skill. The positive effect of the holiday emission scaling would mean $R < 1$, whereas $R > 1$ indicates that the actual emission moved into the opposite direction of that suggested by the Sunday scaling coefficients.

## 4 Results

### 4.1 Overall short-term impact of public holidays

The summary of the simulations is presented in Fig. 2 TS8 for the main holidays of 2018 and all considered pollutants. The physical meaning of the $R$ criterion (Eq. 1 TS9) is illustrated in Fig. 3, which shows a substantial "jump" in all model skills at or around Christmas Day. Before and after that day the skill values are similar. The HS run exhibits less of a jump than the BL case, which indicates that the model–measurement agreement is more homogeneous. The ratio of the standard deviations of the skills $R$ from Eq. (1) is presented in Fig. 2 for all skills and all species.

The effect, expectedly, varies between the quality metrics and species. Thus, the least sensitive parameter is RMSE, whereas the spatial correlation coefficient showed mixed signals in loose connection with other parameters. The most sensitive parameters are bias, fractional bias, and fractional gross error, which are also the most important for the study.

The majority of metrics and cases showed a clear positive effect of accommodating the holiday emission changes in the model simulations. The most significant changes were obtained for $NO_2$ and $NO_x$, where the flattening of, e.g., fractional bias time series could be as large as 10 %–20 %. It reflects the major role of the changes in the traffic intensity (mostly, reduction) during holidays. Carbon monoxide generally followed the $NO_x$ patterns but with a lower effect due to a large background level and the contribution from the sources with weak or no weekly variation in the intensity. Changes in $O_3$ and $SO_2$ were very limited, except for Christ-

mas when they also showed a more homogeneous bias of the HS run.

Intriguingly, the effect for $PM_{2.5}$ and $PM_{10}$ was significant for fractional bias and fractional gross error (but small for bias) and partly detrimental. It indicates that the Sunday profiles for primary PM and, possibly, $NH_3$ emission may be not suitable for holidays. Domestic activities, seemingly adding little to $NO_x$ emissions, may be quite significant for emission of PM and PM precursors. It was particularly evident for May Day, which is usually characterized by intense outdoor activities all over Europe.

Holiday-wise, the most significant impact was obtained for Christmas, while Easter and Ramadan (assessed for Turkish stations only) showed moderate improvement. May Day showed the mixed signal mentioned above.

### 4.2 Examples of specific holidays

The impact of holidays on the SILAM spatial skills was the largest for the Christmas week (Figs. 2 and 3). As expected, the Christmas period is characterized by lower emissions, which resulted in a high bias of the BL model run and almost 50 % growth of the RMSE compared to the surrounding days. The reduction in emission in the HS run improved the performance but did not eliminate the problem: the time series of the skills still exhibit strong jumps on (and around) Christmas Day. Comparison of daily-mean concentrations showed a reduction in the model bias for the HS run by $\sim 4.5\,\mu\text{g}\,\text{m}^{-3}$ of $NO_2$. Consequently, the RMSE was also lower, by $\sim 4\,\mu\text{g}\,\text{m}^{-3}$. These improvements constitute about 26 % of the baseline statistics (see Figs. S1–S6 in the Supplement for other species). However, as seen from the bias time series (Fig. 3), the HS run, being a step in the right direction, incorporated only a small fraction of the actual emission reduction, which also started before and ended after Christmas Day.

Comparing the HS and BL runs for Easter (Fig. 4), one can see a substantial improvement of the scores for the days of the event. Similarly to the winter holiday week, Easter emission reduction seems to be greater than that of Sundays but the difference is not so large (see results for other species Figs. S7–S12).

The first 10 d of May were regarded as an example of late-spring/summer vacations (there are no holidays that apply to all of Europe during summer itself). The HS run showed slightly lower values for RMSE but, similarly to Easter, initially negative bias increased further. Nevertheless, the bias time series became smoother compared to the BL one, which is an indication of the improvement: the systematic emission underestimation is a separate task, the necessity of which should not be masked by another error. Reduction in $NO_x$ resulted in a substantial improvement of the ozone scores (Figs. S13–S18). This connection was the strongest among all holidays throughout the year, owing to the active chemistry and photolysis in May.

| (SD HS/SD BL) | | | | | | | |
| --- | --- | --- | --- | --- | --- | --- | --- |
| **May Day (Europe),** (13 days, 29 Apr–12 May) | | | | | | | |
| | $O_3$ | $NO_2$ | $PM_{10}$ | $PM_{2.5}$ | $SO_2$ | CO | $NO_x$ |
| R_RMSE | 1.00 | 1.00 | 1.00 | 1.00 | 1.00 | 0.98 | 1.00 |
| R_corr | 1.05 | 0.97 | 1.02 | 0.97 | 1.00 | 1.04 | 0.97 |
| R_bias | 1.00 | 0.96 | 1.02 | 1.00 | 1.00 | 0.97 | 0.99 |
| R_FracB | 1.02 | 0.82 | 1.09 | 1.12 | 1.26 | 1.01 | 0.86 |
| R_FGerr | 1.01 | 0.86 | 1.13 | 1.11 | 1.22 | 0.97 | 0.88 |
| **Christmas (Europe),** (12 days, 19 Dec–31 Dec) | | | | | | | |
| R_RMSE | 1.00 | 1.00 | 1.00 | 1.00 | 1.00 | 0.99 | 0.98 |
| R_corr | 1.01 | 0.92 | 1.00 | 0.99 | 1.00 | 1.01 | 1.04 |
| R_bias | 0.95 | 0.97 | 0.99 | 1.00 | 1.00 | 0.94 | 0.93 |
| R_FracB | 0.84 | 0.87 | 0.93 | 0.89 | 0.87 | 0.84 | 0.86 |
| R_FGerr | 1.00 | 0.97 | 0.84 | 0.91 | 0.97 | 0.85 | 0.93 |
| **Easter (Europe),** (9 days, 28 Mar–6 Apr) | | | | | | | |
| R_RMSE | 1.00 | 1.00 | 1.00 | 1.00 | 1.00 | 0.97 | 0.99 |
| R_corr | 1.01 | 0.95 | 1.00 | 1.03 | 1.02 | 1.03 | 1.01 |
| R_bias | 0.99 | 0.98 | 0.99 | 1.00 | 1.00 | 0.97 | 0.97 |
| R_FracB | 0.98 | 0.87 | 0.97 | 0.95 | 1.00 | 0.96 | 0.89 |
| R_FGerr | 1.04 | 0.88 | 1.02 | 1.02 | 1.01 | 1.05 | 0.89 |
| **Ramadan (Turkey),** (33 days, 16 June–18 July) | | | | | | | |
| R_RMSE | 1.00 | 1.00 | 1.00 | 1.00 | 1.00 | 0.97 | 0.99 |
| R_corr | 1.01 | 0.95 | 1.00 | 1.03 | 1.02 | 1.03 | 1.01 |
| R_bias | 0.99 | 0.98 | 0.99 | 1.00 | 1.00 | 0.97 | 0.97 |
| R_FracB | 0.98 | 0.87 | 0.97 | 0.95 | 1.00 | 0.96 | 0.89 |
| R_FGerr | 1.04 | 0.88 | 1.02 | 1.02 | 1.01 | 1.05 | 0.89 |

**Figure 2.** Summary of the *R* value for the main European holidays in 2018 for the considered air pollutants (the effect of Ramadan is assessed for Turkey only).

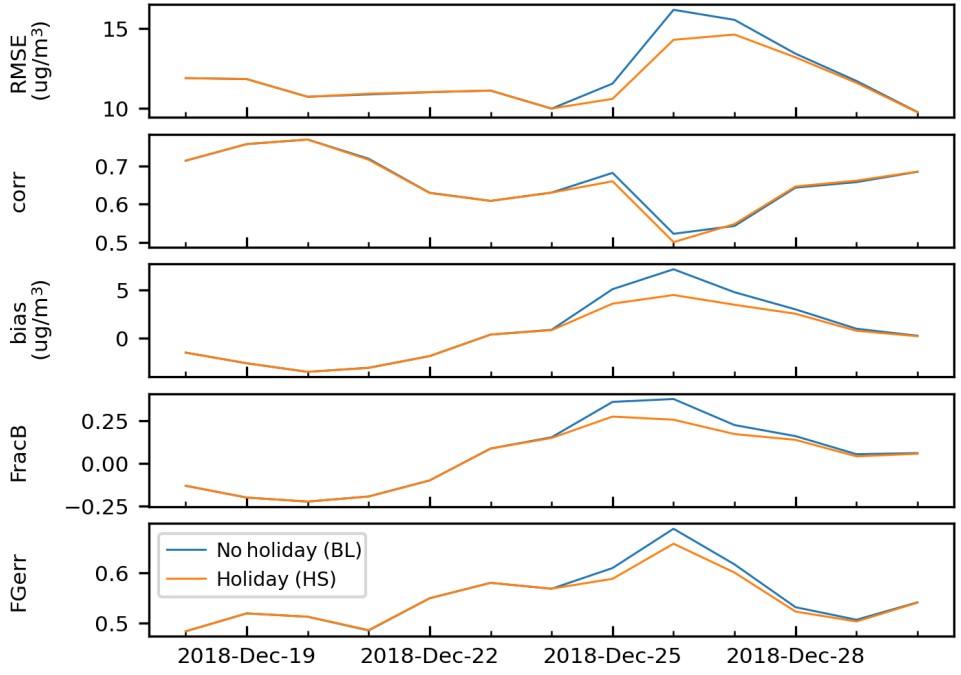

**Figure 3.** SILAM daily-mean spatial scores for Christmas ($NO_2$, all of Europe).

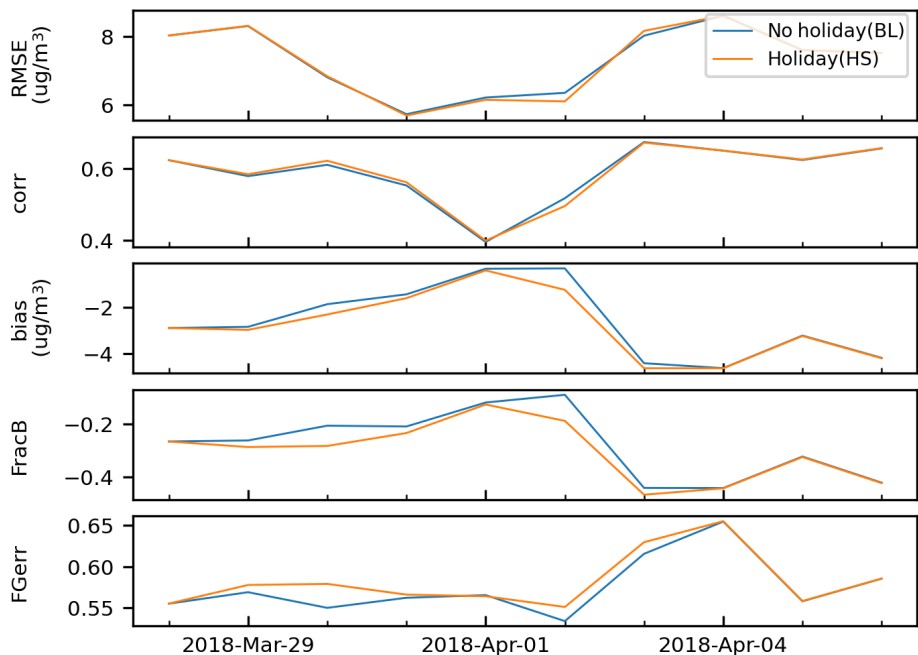

**Figure 4.** SILAM daily-mean spatial scores for Easter (NO$_2$, all of Europe).

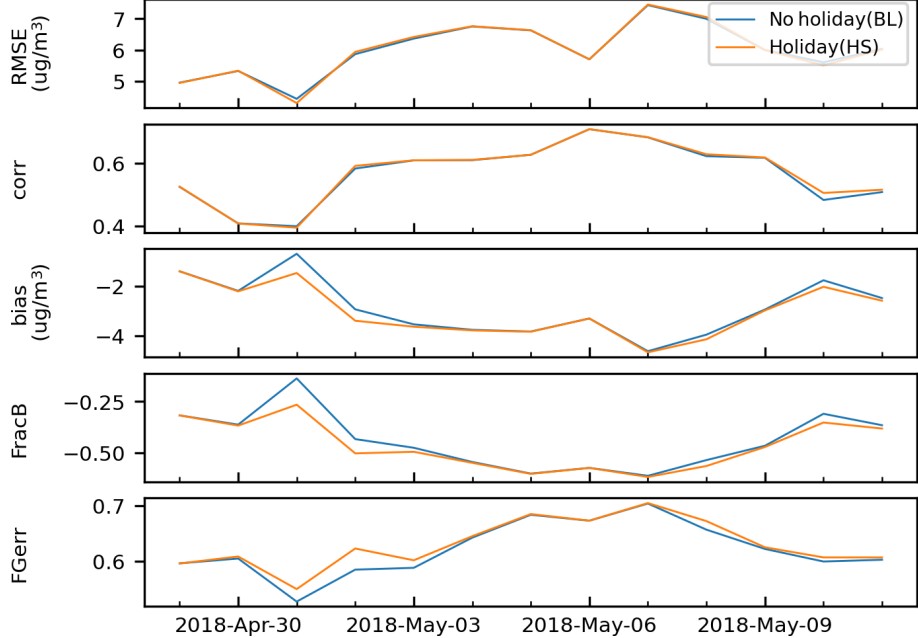

**Figure 5.** SILAM daily-mean spatial scores for May vacations (NO$_2$, all of Europe).

In the Muslim countries (Turkey, Albania), the Ramadan month is not a public holiday; only working hours are reduced, which is not reflected in the HS run. Only the last 3 d of Ramadan – the Ramadan Feast – are public holidays in Turkey (Table 3, Figs. 2 and 6 for NO$_2$, Figs. S19–S24 for other species). For these days, there are distinct differences between the BL and HS model runs. However, similarly to Easter and May Day, the model is generally low biased for NO$_2$ in Turkey during this period; therefore the additional reduction in the concentrations is, formally speaking, not an improvement: the negative bias increases. Nevertheless, it is a step in the right direction, as seen from the reduced variations in the model skills of the HS run (Fig. 2). Due to this underestimation, it is difficult to say how conser-

vative the Sunday-level emission reduction is for these holidays (Fig. 6).

Unlike the Christmas and Easter holidays, which exist in most European countries, the Ramadan Feast days only affect Turkish stations substantially. At the European scale, the effect is negligible.

## 4.3 Long-term statistics

At the annual scale, the impact of holidays on the model performance is limited. The reduction affects only the days with changed emissions and practically does not influence even the next day. The most significant impact was for the Christmas and New Year weeks, but even for them the effect faded out by the next day. According to the annual statistics, at an annual level the overall effect for $NO_2$ for all of Europe was positive but did not exceed 1 %, which reflects the typical number of holiday days in a year (< 3 %) and up to ∼ 30 % improvement during these days. The impact on other species was lower than that for $NO_2$.

## 5 Discussion

### 5.1 Impact of holiday effect on model skills: episodically significant, noticeable at an annual level

The simulations presented in the previous section confirmed that the official holidays substantially affect air quality, as also shown in the studies outlined in the Introduction. The holiday incorporation into the simulations as Sundays, being very simple technically, brings noticeable improvement of the model skills for the days with the modified emission. Since the number of such days in each year is < 3 %, the overall improvement of the annual skills is within 1 %, which is quite significant at such a level of aggregation.

The suggested simple approach should be regarded as only the first step. Holidays are characterized by redistribution of emission due to changing traffic structure, a shift in activities from office areas to suburbs, etc. Incorporation of these effects can further improve the model skills but will require quantitative information on such redistribution at the European level. Several approaches towards determining these profiles have been reported (e.g. Guevara et al., 2021; Mues et al., 2014; Menut et al., 2012), but tests with SILAM showed no substantial improvement suggesting additional uncertainties in the proposed profiles. Some support can be found from traffic information, which is presently not available at continental scales (examples for two cities are provided below).

### 5.2 Sunday-based emission reduction for holidays is a conservative estimate

The simulations also suggested a comparatively simple way to achieve a more significant gain: the Sunday emission scaling (Fig. 1) can be amplified. In a few cases, especially for Christmas and New Year, the actual emission rates might be much lower, whereas for some events the emission of some species might increase. Thus, the New Year's Eve celebration in many countries involves fireworks, which add substantial amounts of PM. The second issue is that the Sunday diurnal profile of traffic (also other sources) is substantially different from that of the weekdays. In the present version of SILAM this difference is not accounted for, which evidently limits the model performance and the gain due to the holiday incorporation.

This is consistent with the estimates of the observation-based studies. Thus, Hua et al. (2021) also found that the holiday effect is much stronger than the weekend effects. They noticed the opposite signs for $PM_{2.5}$ and $NO_2$: an average increase of about 22 % and an average decrease of about 11 %, respectively. Similarly, Retama et al. (2019) reported a substantial effect of fireworks on PM at night and the following morning of Christmas Day and New Year's Day. Along the same lines, Rozbicka and Rozbicki (2016), demonstrated that daily mean ozone concentration and maximum ozone peaks are, respectively, 13 % and 8 % higher than those on the weekdays, which also indicates a reduction in $NO_2$ concentrations of about 20 %. Conversely, the Nodehi et al. (2018) study showed that the Nowroz CE5 holidays (the Iranian New Year or spring festival) are characterized by a reduction in concentration of $PM_{2.5}$ due to the reduction in the working activities and no massive fireworks. The reported reduction in $PM_{2.5}$ concentration during the Ramadan Feast holidays is quite close to our estimates.

### 5.3 Regional specifics of the effect of HS and R3 emission reduction

The impact of holiday-related emission reduction varies from country to country with substantial differences visible even at a sub-country level. To highlight these peculiarities, we used the station-wise temporal correlation coefficients for hourly $NO_2$, CO, $O_3$, and $PM_{2.5}$ concentrations (Figs. 7–9). The maps reveal a strong inhomogeneity of the effect for the Christmas and New Year weeks (Fig. 7) as well as for May Day (Fig. 8). It can dramatically vary even within a single country – as seen from the comparison of maps in Fig. 7 and the country-median correlation coefficient of Fig. 9.

In the case of $NO_2$, correlation increases, e.g. in northern Germany, Italy, Poland, and the eastern part of Finland for both the HS and the R3 runs: reduction in emission had led to lower concentrations, which improved temporal correlation for these regions. Conversely, there was no effect or

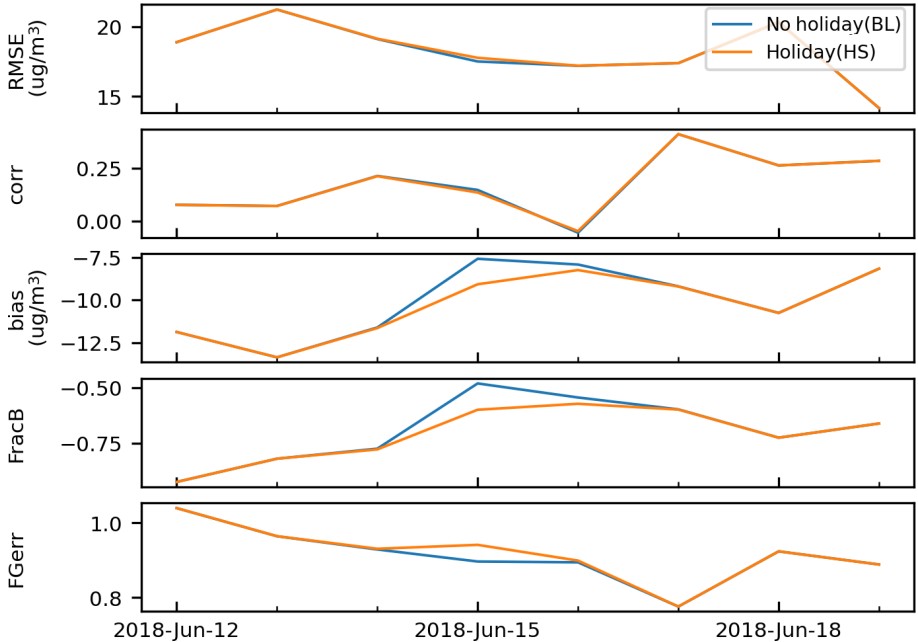

**Figure 6.** SILAM daily-mean spatial scores for Ramadan ($NO_2$, only stations in Turkey).

even deterioration of correlation in southern Germany, northern France, the Madrid region, etc.

Other species showed qualitatively similar patterns but lower gains and losses. Significant changes are noticeable only for CO, which is also significantly affected by traffic. Minor changes for ozone were noticeable only in winter when $NO_x$ emissions affect $O_3$ concentrations via titration. This is consistent with the spatial statistics of Fig. 2. For PM, the effect was not unequivocal: there is a small but coherent reduction in correlation in eastern Europe in May but a neutral response or an increase for Christmas. This once again refers to the regional habits of celebrating these holidays and corroborates the overall detrimental effect on these species reported in Fig. 2. One should also keep in mind that the fireworks are used during the New Year celebration only in some countries (as suggested by the current results, in western Europe), where the HS and R3 runs are clearly inadequate for PM.

Surprisingly, for the Christmas holidays, skills over most of France are generally worse for the HS run and much worse for R3 indicating a substantially different pattern of activities during holidays, compared to those of the neighbouring countries: reduction in $NO_x$ emissions and, consequently, concentrations there do not correlate with the observed tendencies. For May Day, the specificity did not show up: correlation has noticeably increased over most of the country, similar to its neighbours. Among the hypothetical reasons for such behaviour, one could suggest more "active" habits for Christmas celebration in France than in the neighbouring countries.

The R3 run, which was planned as an overshot, showed strong improvement of temporal correlation over Christmas week in eastern Europe, central and northern Italy, and northern Germany. Therefore, one can argue that even the 5-fold emission reduction in these countries/regions might be not that much of an exaggeration.

These issues deserve a more detailed analysis accounting for the varying traffic patterns and effects on days preceding and following the official holidays.

### 5.4  Local traffic counts illustrate the phenomenon

As mentioned above, a lack of systematic continental-scale traffic counts precludes their usage for determining and/or verifying the assumptions of the current study. However, for a few cities the data are available and can be used as an illustration of the effect. Below we provide the time series for Helsinki and Dublin (Fig. 10). The daily traffic counts over several years corroborate/illustrate the above discussion. Indeed, for Helsinki, the May Day traffic count almost perfectly meets the Sunday number of cars 3 d before in 2019 and 1 d after in 2020. The difference between the years illustrates the COVID-19 lockdown effects in 2020.

For Dublin, the Christmas–New Year holidays for 2 sequential years show that for this major event the traffic reduction is at least 2 times greater than for an ordinary Sunday: almost 4 times less cars were counted on 25–26 December than on an ordinary day. Such a reduction is already comparable to the 5-fold reduction in the S3 run. The city also manifests about twice lower traffic intensity during COVID-19 lockdowns, whereas Helsinki lost about 30 % of its traffic.

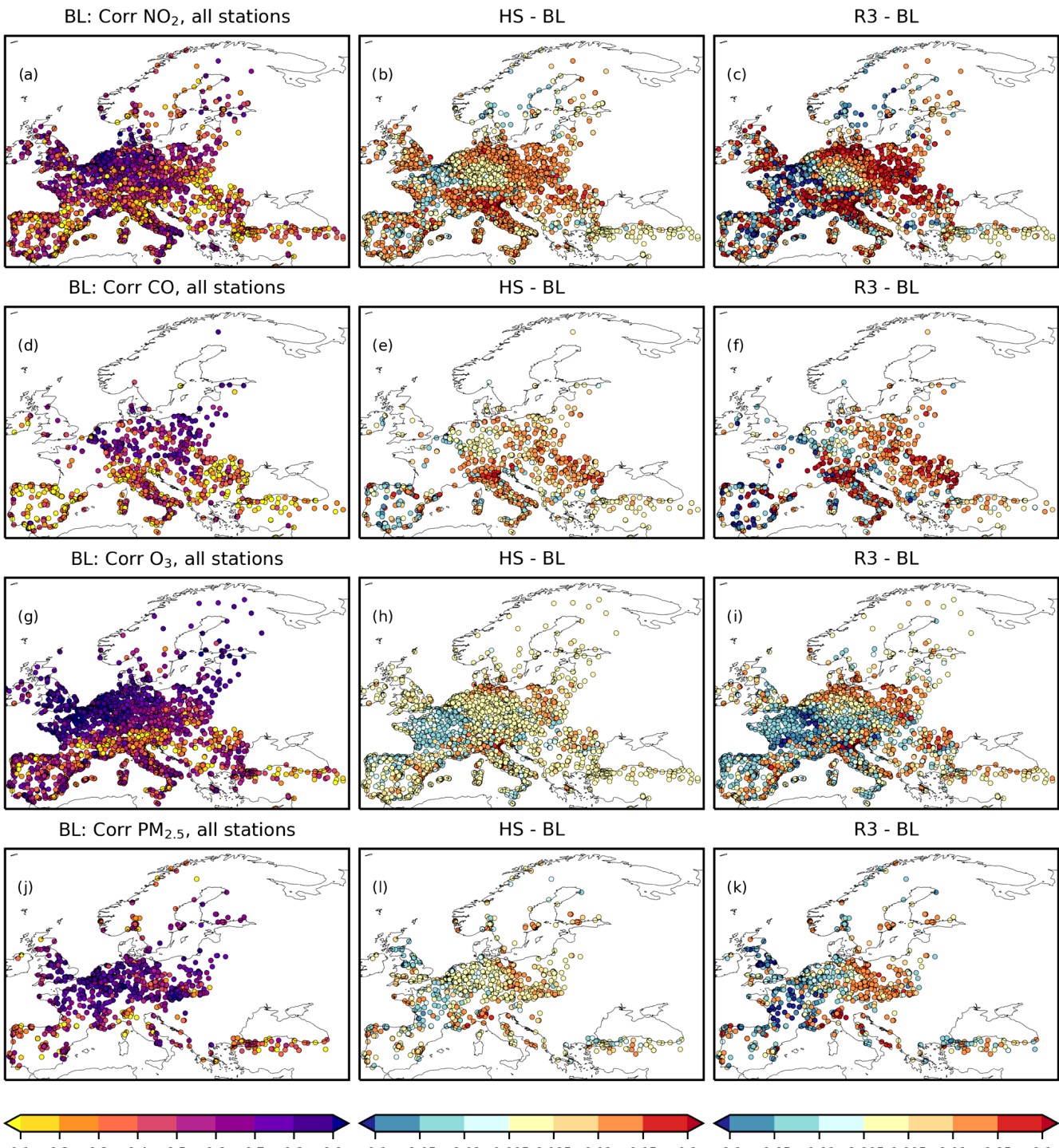

**Figure 7.** Maps of the temporal correlation coefficient of hourly $NO_2$, CO, $O_3$, and $PM_{2.5}$ concentrations for the EEA stations during the Christmas holidays (21–31 December 2018). CE6

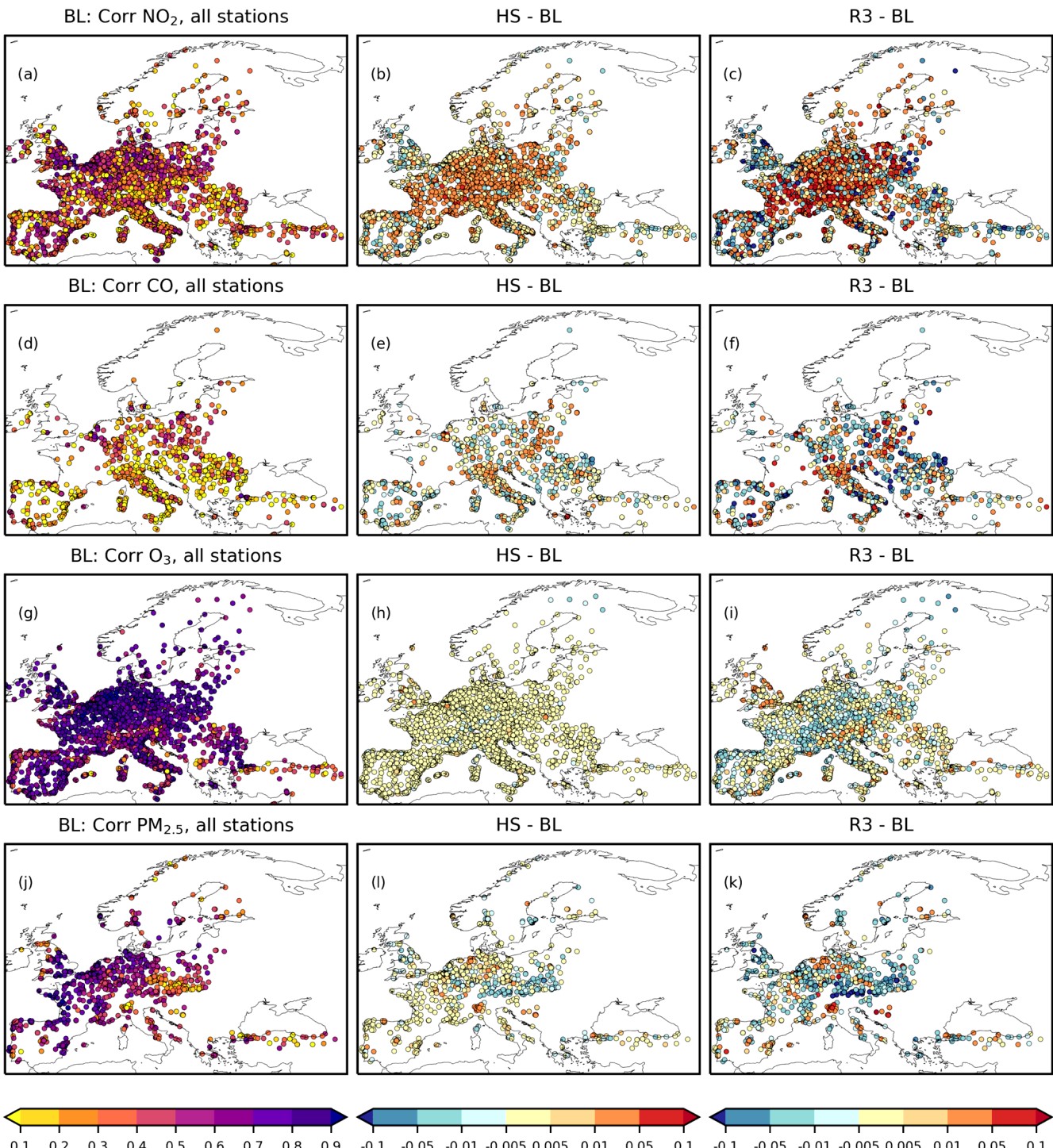

**Figure 8.** Maps of the temporal correlation coefficient of hourly $NO_2$, $CO$, $O_3$, and $PM_{2.5}$ concentrations for the EEA stations during the May Day holidays (29 April–11 May 2018).

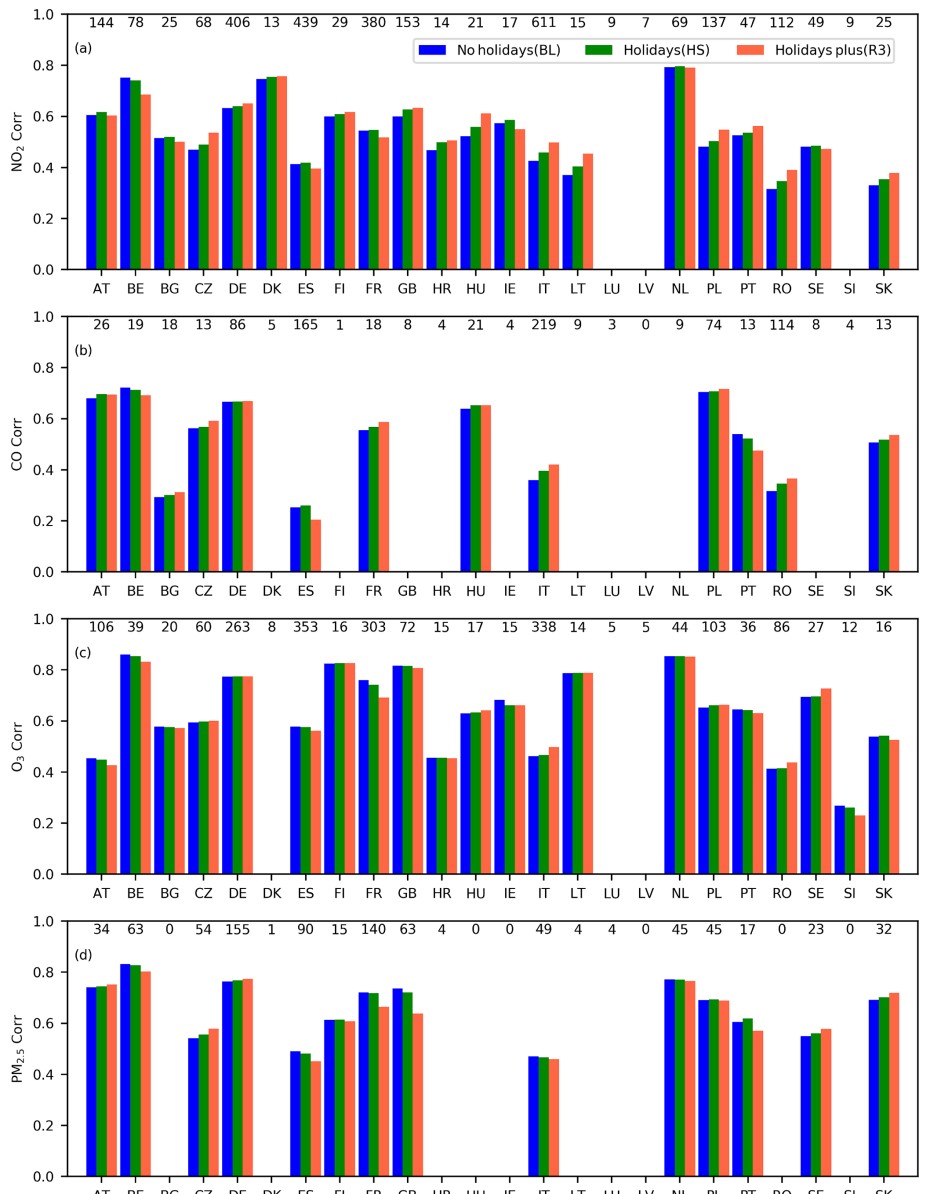

**Figure 9.** Country-wise median change in temporal correlation coefficient during the 2 weeks of Christmas holidays (21–31 December 2018). The numbers at the top of each panel show the number of stations that reported data for the period.

Finally, one can see that the traffic is not restored to normal intensity between Christmas and New Year, similar to what was noticed from the observations (Fig. 3).

## 6  Summary

Incorporating information on public holidays in emissions of the affected anthropogenic sectors leads to substantial short-term improvements of the SILAM model scores, even if done conservatively. The largest impact was found for $NO_x$, which is controlled by the changes in the traffic intensity. Certain improvements were also found for other species, but the signal was weaker than that for $NO_x$.

The effect of the emission reduction during holidays may look detrimental in the case of a systematic underestimation in some regions. However, in the majority of such cases the bias and other skills became more homogeneous in time manifesting a reduction in the holiday-induced errors in emission.

The sensitivity runs confirmed that the Sunday emission level, in many cases, is too conservative a proxy for the public-holiday emission. Thus, the reduction during the Christmas and New Year holidays of 2018 was closer to a

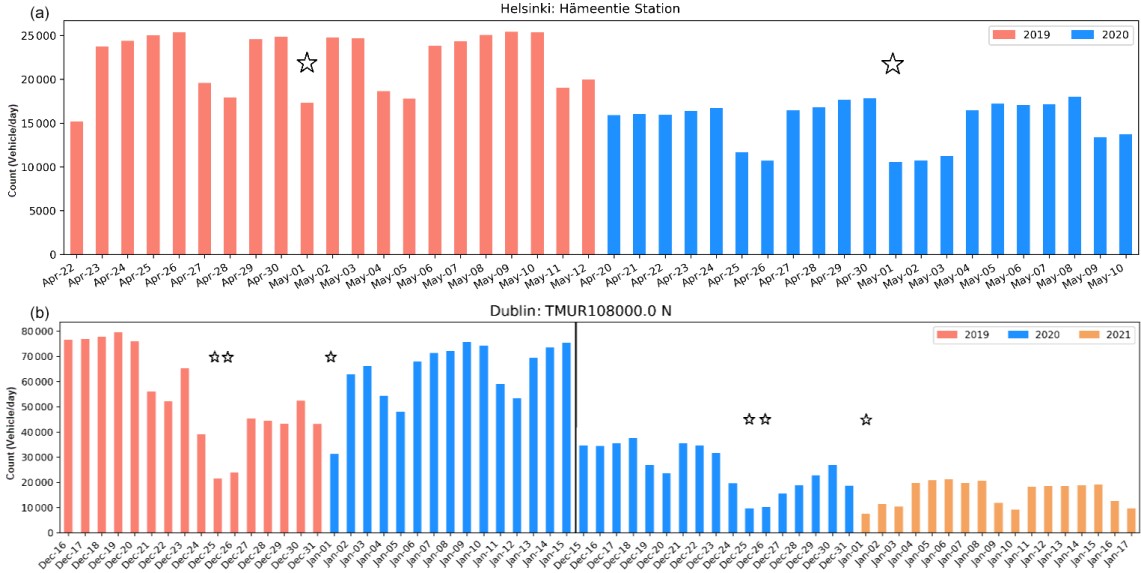

**Figure 10.** Daily traffic count in Helsinki **(a)** and Dublin **(b)** during Christmas/New Year and May Day holidays. Stars mark the official holidays. Obs non-holiday day of 6 January in Dublin CE7.

factor of 4 in western Europe and possibly even stronger in eastern Europe.

The current experiment used the prescribed sector-specific diurnal profiles of emission intensity, which were the CE8 same for weekdays, weekends, and holidays. The incorporation of specific profiles for weekends and holidays might further improve the quality of the model predictions.

The proposed method of handling emission reduction in AQ models, albeit very simple and with a room for improvement, gives noticeable gains in the model performance. The method is straightforward to implement in the AQ models and can be regarded as an easy way to improve the model prediction skills for the periods of public holidays. An in-depth analysis of the specific holidays and related traditions in specific countries, such as fireworks on New Year's Eve, would, most probably, lead to further improvements of the AQ predictions.

*Code and data availability.* SILAM is an open-code system and can be obtained from the GitHub open repository: https://github.com/fmidev/silam-model (last access: TS10). The simulation results are available on request from the authors of the paper.

*Supplement.* The supplement related to this article is available online at: https://doi.org/10.5194/gmd-14-1-2021-supplement.

*Author contributions.* The authors jointly devised the project and developed the paper concept. YF contributed to the implementation of the research and analysis of the results and drafted the paper. RK performed the SILAM computations and contributed to the analysis. MS contributed to the analysis, drafted the Discussion section, and contributed to other sections of the paper. All authors edited the final text.

*Competing interests.* The authors declare that they have no conflict of interest. TS11.

*Acknowledgements.* The study was performed within the scope of the Academy of Finland project GLORIA (grant no. 310372). Financial support of the Copernicus Atmospheric Monitoring Service (CAMS-50 and CAMS-61) for the SILAM development is kindly acknowledged.

*Financial support.* This research has been supported by the NAME OF FUNDER (grant no. GRANT AGREEMENT NO). TS12

*Review statement.* This paper was edited by Jason Williams and reviewed by three anonymous referees.

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

TS37  Please provide the publishing journal.

TS38  Please provide the publishing journal.

TS39  Please provide the publishing journal.

TS40  Please provide the publisher and place of publication (city).