# Peer review of "Effect of accounting for public holidays on skills of atmospheric composition model SILAM v.5.7"

_Geoscientific Model Development, 2021_

## Author Response (AR1)

Dear Editor,

Please see the enclosed revised version of the paper by Fatahi et al "**Effect of accounting for public holidays on skills of atmospheric composition model SILAM v.5.7**". We would like to thank the reviewers and the editorial board for their criticism and suggestions, which we followed while preparing the revised version. Below, we provide point-by-point responses to the comments.

**Authors responses to Reviewer:**

**Reviewer #1:**

**General comment:**

This paper is a positive contribution towards the improvement of regional CTMs as it outlines the limits of a simple method that involves modulating emissions during public holidays and paves the way for further model improvements.

Authors argue that the proposed method of handling emission reductions provides positive gains in model performance scores as well as that the method "can be considered as an easy way to significantly improve the model prediction skills". However, these are bold conclusions that can not be easily supported based on the results presented in the preprint. E.g. it turns out that the improvements in the correlation coefficients are very inhomogeneous across Europe (at least some countries like France appear to be particularly problematic with no suggested hypothesis as to why this is the case) and benefit in terms of bias in many cases relies on future improvements in emission inventories. A more veritable outlook would be that the results presented, highlight the potential for improving modelling skill by providing valuable insights into when, where and how, simple, targeted emission modulations benefit models. However, as mentioned by the authors, further in-depth analysis will be required to evolve the method in a way that more consistent results can't be obtained both spatio-temporaly and across evaluation metrics and thus render it appealing for general use.

**Authors Responses:** Thank you for the analysis! Indeed, the improvement was inhomogeneous over Europe and varied between holidays. However, the continental-scale skills went up, which justified the approach. Also, the correlation coefficient for NO2 and, to a less extent, CO in such regions as Eastern Europe, Italy, Spain, UK etc. went up by 0.05 – 0.1, which is a very substantial improvement for these

areas where the baseline model skills are not impressive. This is made clear in the revised Discussion. The issue of some countries / regions benefiting less than others also receives more attention. The revised version of the paper now includes the Figure 9, which complements the Figure 8 for May Day. These figures contain full evaluation for all stations for the corresponding periods. Concerning the overall message, which we tried to clarify in the revision, our goal was to suggest something very simple, which could improve the skills prior to lengthy studies, pick the "low-hanging fruit" and highlight the areas for development. One of such areas is certainly the French specifics in Christmas - and until this paper, we were not even aware about it (in terms of absolute SILAM skills, France is not different from its neighbors). The revised version now includes the other holiday period – 1 May, where France is by no means different from its neighbors. It is now highlighted in the Discussion.

**Specific comments:**

- Figures 2-5, each include two stations in the Netherlands (not the same ones in all figures). Proper names of the stations (i.e. locations) should also be included rather than the cryptic NL codes, as well as the station type (rural, urban-background etc.). But there are more reasons for concern here. Understandably, these figures can not accommodate a multitude of time series from European stations, however there is no justification as to why the Netherlands provides good enough examples for time series during the examined periods, nor how these particular stations were chosen. A more fundamental concern would be that concentrating only in the Netherlands conveys a partial outlook of the effect as far as time-series are concerned, thus also hindering better understanding of the impact of the holiday emission reductions. For example, the Netherlands is not particularly known for it's Easter time festivities (figure 4), nor the May vacation happens at the exact same time across the country, thus putting into question the usefulness and certainly complicating the interpretation of figure 5 (b) and (c). Please consider using also stations from other countries with different characteristics (geographical, cultural etc.), by also providing some justification of the selection criteria.

**Authors Responses:** The stations in the Netherlands were chosen because the model skills there (in particular, bias and correlation) are among the best over the whole continent (see e.g. Fig.8, left column, for Christmas). It allowed direct attribution of the signal to emission rather than to model deficiency. Secondly, the Netherlands is one of the most-polluted places in Europe with very strong traffic contribution – the one strongly affected by holidays. And $NO_2$ is the primary pollutant directly emitted by traffic. The revised paper gives more attention to other species but still, the strongest signal is visible

for NO2 in the areas with the highest traffic intensity. The stations details are now included in the captions. The station selection was quite random – Netherlands is a small country and SILAM performance is quite homogeneous across its area. This is now explained in Evaluation section.

- Paragraph 5.2: The discussion on the Figure 7 relies heavily on the claim that the performance of the model is "very good", but no accompanying evidence is presented to support that the performance is equally good in all NO2 concentration regimes. After the 24th of December, the air quality situation apparently changes and both model (BL) and observations acquire generally higher values. Can reasons other than emissions be ruled out (e.g. meteorology) or is such an increase in concentrations expected already in the BL case due to increased (!) holiday activity? In essence, does the model perform equally well (e.g. in terms of bias) in those higher concentration conditions so that we can we reliably attribute differences between observations and model (BL) to real emission changes? If not, a possible systematic bias in the model in these different conditions could be entangled with the holiday emission reduction effects thus challenging the presented interpretation. Please consider including a different station with no such pronounced jump in NO2 levels or support better the claim that the model performs equally well in such conditions.

**Authors Responses:**

Thank you for pointing out at this dynamic! However, the reason seems to be somewhat different. The rise of concentrations on 25 December is _mainly_ due to meteorological conditions: the emission in the model did not change but the predicted concentrations grew, in good correspondence with the observations. Then meteorology is the only parameter to blame. However, the holiday period itself lasted much longer than the formal day off, from 26 December practically till New Year. The HS and R3 reductions were limited to 26 December whereas observations suggest ~60% lower emission until the end of the year. Noteworthy, model suggested that the days 26-30 December are characterized by a similar level of pollution with a slight upward tendency, which is well in agreement with measurements. We expanded the corresponding discussion to make it clear. We refined the statement on the model skills at that particular station stressing that the "very good agreement" refers to the specific episode (actually, it is good also for other periods but it is not relevant for the specific discussion).

**Technical corrections:**

- Line 53: Milan

Authors Responses: Corrected

- Line 254: overshoot

Authors Responses: Overshot seems correct in that place.

- Please check Grivas et al. reference, not evident where this was published.

Authors Responses: The journal name is Atmosphere

**Reviewer #2:**

This manuscript presents and evaluates an approach to incorporate the effect of public holidays on European air quality models. The methodology consists on a simple technic based on the scaling of primary emissions at the country level when holidays occur. Two hypotheses are tested: a first one in which the levels of emissions during holidays are assumed to be equal to the Sundays ones, and a second one where an 80% emission reduction is considered during holidays. Both approaches are evaluated by comparing modelled results against in-situ observations. As described by the authors in the introduction section, which I find it very detailed and comprehensive, several observations-based studies have already highlighted the effect that specific holidays can have on pollutant concentrations. However, a systematic analysis of the holidays effect from a modelling perspective at the EU scale, as well as the description of a methodology to properly incorporate it into AQ models has not yet been addressed with the same level of detail. Therefore, the topic presented in this manuscript is of interest and represents a good contribution to GMD. Nevertheless, there are several aspects of the manuscript – including methods, evaluation and discussion of the results – that, in my view, are not sufficiently convincing in their current form and should be carefully revised before the manuscript can be accepted for publication.

**Major comments:**

C1. In one of the sensitivity tests, authors assume an 80% emissions reductions during holidays. Authors already mention in the manuscript that the presented approach should be considered only as a first step. However, I think that the hypothesis made (80% reduction for all sectors considered) should be backed up by the analysis of sectoral related activity data. While it is true that for certain sectors this analysis can be challenging due to the lack of data, for others there is information that can be used for this purpose. In the case of the A_Public Power industry sector, authors could use the ENTSO-E transparency platform (https://transparency.entsoe.eu/), which reports data on hourly electricity generation by fuel type per country. In the case of F_Road Transport, authors could use information on traffic counts reported by

national transport agencies, such as the Finish transport agency (https://vayla.fi/en/transport-network/data/open-data/road-network/tms-data).

**Authors Responses:** Thank you for the suggestion. The changes of daily traffic volume in Helsinki and Dublin were examined in recent years and two charts have been added to the revised manuscript Discussion. Concerning the 80% reduction run, we made it as a clear overshot, to estimate the maximum (un)feasible effect and to assess the lowest boundary of the changes. But we did not change sectors, which do not have weekend decrease in the emission inventories

C2. Following with the previous comment, it is questionable that all emissions from the C_OtherStationaryCombustion sector suffer an 80% reduction during holidays. In the case of PM, between 80 and 90% of total emissions are related to residential wood combustion activities (https://www.ceip.at/). Several studies have shown that residential wood combustion activities in Europe tend to increase significantly during weekends when compared to weekdays, as people use this fuel for recreational purposes. Examples of these studies are Krecl et al. (2008) and Athanasopoulou et al. (2017). I would expect a similar behavior during holidays (specially Christmas period), when people spend more of their time at home. The PM10 and PM2.5 spatial scores for Christmas shown in the supplementary material already suggest that with the holiday days considered as Sundays (the HS case) the skills of the model deteriorate (correlation decreases and MB increases). Authors suggest that this could be related to the use of fireworks, which are not accounted in the CAMS emission inventory, but this is not proved in the manuscript.

**Authors Responses:** You have raised an important point here. We agree with this complexity and also pointed it out in the Introduction. However, both quoted papers refer to the weekend emission profiles, i.e. they, at least in theory, should be already included in the GNFR emission temporal profiles. Therefore, our HS run would take them into account automatically. The R3 run, indeed, reduces also that sector but it was not planned as a realistic exercise – rather as a definite low-boundary. Accordingly, the Materials and Discussion sections were revised to emphasize this point.

C3. In the abstract section, authors mention that "Spatial and temporal distributions of atmospheric concentrations of the major air pollutants (PM2.5, PM10, SO2, CO, NO2, NOX, and O3) were

considered". However, the analysis, evaluation and discussion of the results is very much focused on NO2. Figures 2 to 7 show NO2 results, while results for CO, O3 and PM2.5 are only shown on Figures 8 and 9 (no results for SO2 or PM10 appear in the main manuscript, only in the supplementary material). A more balanced discussion of all the pollutants considered should be provided (or, alternatively, the pollutants not shown in the main manuscript could be removed from the study). In the case of O3, the discussion is focused on the Christmas period, when O3 levels are very low. Discussions for this pollutant should be focused on Easter. In the case of NO2, time series are shown almost exclusively for stations in the Netherlands (Figures 3,4,5 and 7). Considering that the study is performed at the EU level, it would be interesting to see specific results in other regions.

**Authors Responses:** The Abstract has been revised to highlight the main stress of the paper. Our analysis shows that NO2 is the most-sensitive pollutant to the weekend and holiday days, so we maintained the primary attention in this direction. The text of Discussion was revised to balance it somewhat and also to show the effect on other pollutants.

C4. The same prescribed sector-specific emission diurnal profiles are used for weekdays, weekends and holidays, which is a limitation of the study. Authors mention several times in the text that the incorporation of specific weekend and holidays diurnal profiles should be done when available. However, several works have already reported in the past specific Saturday/Sunday diurnal profiles for the road transport sector, which is the main contributor to total NOx emissions. Examples of these profiles can be found in Pregger et al., (2007); Menut et al. (2012); Mues et al. (2014) and Guevara et al. (2021), among others. Following with the hypothesis made by the authors at the weekly level, the same diurnal profiles proposed for Sundays could be assumed for holidays, at least for the road transport sector. I think this point should be addressed more carefully, and perhaps it would be good to produce an extra AQ run assuming a set of specific weekend/holidays diurnal profiles. This would, for sure, bring an added value to the study.

**Authors Responses:** Indeed, several studies have suggested diurnal profiles for the weekends, we are aware of those papers, participated in the evaluation of the Tempo profiles of Guevara et al, etc. The problem however was that the model did not gain much of skills when using these profiles, so they were not introduced. Our current hypothesis is that the diurnal profiles should be highly specific to country /

region / season to provide noticeable benefit. Since this is a separate topic only partially affecting the current study, we preferred to put it as a future research needs. This is now clarified in Discussion.

Ps: At the last meeting, we agreed to dig into suggested references…..to answer this comment…..

**Other comments:**

1.  In section 2.1 authors mention that they consider events marked with "National holiday", "local holiday" and "common local holiday" when retrieving holiday events from the Calendarific API. I understand that "local holiday" and "common local holiday" refer to holidays that are only occurring in a specific region(s) of the country – while in the rest of the country is a normal working day. Considering that the emission scaling approach proposed is at the country-level, should not only "National holiday" be considered?

    **Authors Responses:**

Multiple types of holidays and observances have been supported at the Calendarific website. The list of the holiday types includes:

*   national - Returns public, federal and bank holidays
*   local - Returns local, regional and state holidays
*   religious - Return religious holidays: buddhism, christian, hinduism, muslim, etc
*   observance - Observance, Seasons, Times

We included the first and the second types as a compromising solution between missing holidays and including regions where particular holiday day is not marked. The total number of holidays of each type is, roughly, 800 vs 100 vs 20 for national/state, regional and local. Therefore, the inevitable error (in either direction) with regional / local holidays is anyway within ~10%. It is now made clear in the revised paper.

2.  In Table 4 – Meteorological driver, should not be "interpolated to 0.2x0.2" instead of "0.1x0.1"?

    **Authors Responses:** Corrected

3.  Line 135, define GNFR acronym + revise the number of GNFR sector (it is 16 and not 7. Note that the GNFR_F sector is split by fuel type)

    **Authors Responses:** Corrected

**References:**

Pregger, T., Scholz, Y. & Friedrich, R. Documentation of the Anthropogenic GHG Emission Data for Europe Provided in the Frame of CarboEurope GHG and CarboEurope IP - Final Report. Stuttgart, http://carboeurope.org/ceip/products/files/Pregger_IER_Final_Report_Feb2007.pdf (2007).

Menut, L., Goussebaile, A., Bessagnet, B., Khvorostiyanov, D., and Ung, A.: Impact of realistic hourly emissions profiles on air pollutants concentrations modelled with CHIMERE, Atmos. Environ., 49, 233–244, https://doi.org/10.1016/j.atmosenv.2011.11.057, 2012.

Mues, A., Kuenen, J., Hendriks, C., Manders, A., Segers, A., Scholz, Y., Hueglin, C., Builtjes, P., and Schaap, M.: Sensitivity of air pollution simulations with LOTOS-EUROS to the temporal distribution of anthropogenic emissions, Atmos. Chem. Phys., 14, 939–955, https://doi.org/10.5194/acp-14-939-2014, 2014.

Guevara, M., Jorba, O., Tena, C., Denier van der Gon, H., Kuenen, J., Elguindi, N., Darras, S., Granier, C., and Pérez García-Pando, C.: Copernicus Atmosphere Monitoring Service TEMPOral profiles (CAMS-TEMPO): global and European emission temporal profile maps for atmospheric chemistry modelling, Earth Syst. Sci. Data, 13, 367–404, https://doi.org/10.5194/essd-13-367-2021, 2021.

Krecl, P., Hedberg Larsson, E., Ström, J., and Johansson, C.: Contribution of residential wood combustion and other sources to hourly winter aerosol in Northern Sweden determined by positive matrix factorization, Atmos. Chem. Phys., 8, 3639–3653, https://doi.org/10.5194/acp-8-3639-2008, 2008.

Athanasopoulou, E., Speyer, O., Brunner, D., Vogel, H., Vogel, B., Mihalopoulos, N., and Gerasopoulos, E.: Changes in domestic heating fuel use in Greece: effects on atmospheric chemistry and radiation, Atmos. Chem. Phys., 17, 10597–10618, https://doi.org/10.5194/acp-17-10597-2017, 2017.

---

## Referee Report (RR1)

The manuscript aims to show that the performances of air modelling systems during holidays are low due to an overestimation of emissions' levels. The study is carried out at European scale with an well-known air quality model (SILAM) and uses the setup of Copernicus Atmospheric Monitoring Service (CAMS) regional air quality forecasts.

The authors propose two ways to reduce emission levels respect to the base case (BL) "with the holiday days considered as Sundays (the HS case)" and "with holidays getting 80% of emission reduction for the sectors affected by the DOW profile (the R3 case)". The impact of these three simulations is shown only for temporal correlation coefficient of hourly $NO_2$, CO, $O_3$, and $PM_{2.5}$ concentrations (Figs.8-10) and for $NO_2$ concentration during Christmas period at one station: NL00107 (Fig.7). Figs. 2-6 show daily statistics for $NO_2$ concentration only for BL and HS simulations during Christmas, New Year, Easter, May and Ramadan holidays.

The purpose of this study to investigate at EU scale, in a systematic way, the effect of the holidays' emissions on air quality models' predictions is interesting, yet the manuscript does not show in comprehensive and concise manner that. The reader would like to see the impact of emissions on $NO_2$, CO, $O_3$, and $PM_{2.5}$ concentrations, at stations and over whole Europe, in the manuscript, not in the supplementary material where specific results at stations can be shown. The study also does not show and discuss the relation between concentrations and emissions' reductions (HS and R3 scenarios) as a whole and by country as a marker for "regional specifics". Moreover, an evaluation of HS and R3 assumptions at stations, by country and station type can give indications about the value of the hypotheses used globally.

In addition to a more careful analysis of the results, their presentation and discussion should be improved, both for language and rationale. Also not all claims are justified. For example "line 185 The impact of holidays on the SILAM spatial skills was the largest for the Christmas week (Figure 2a)". This comment should be supported by Fig.2 where all the holidays should be shown. As it is, this statement seems to be valid for all pollutants.

Moreover, an analysis and discussion of results as a function of pollutant type, supported by data and images should be included in the study. For example, intuitively, it is expected that the impact of reductions $O_3$ during spring will be different from winter.

The research topic under investigation in this study is of interest for air quality modelling community but the manuscript has too many pitfalls in all parts, except Introduction, therefore I would not recommend the manuscript for publication in the current form.

---

## Author Response (AR2)

**Response to referee comments to the paper Fatahi et al "Effect of accounting for public holidays on skills of atmospheric composition model SILAM v.5.7"**

We would like to thank the reviewers for the critical comments to the revised version of the paper. They gave us a fresh look at the manuscript and facilitated its revision. Below we outline the introduced changes and answer to the criticism point-by-point.

**Response to the general criticism**

Following the advice of the Editor and referring to the generic criticism, we substantially reviewed the presentation of the results making them more specific and removing the obscure pieces:

- we introduced a quantitative criterion of the improvement (section Methods), which stressed the purpose of the exercise: to make the model skills, first of all, bias, more homogeneous in time avoiding / reducing their jumps during holidays. This objective differs from the "default" goal of improving the formal model skills. They coincide in cases of the positive model bias when the lower emission in holidays leads to the skill improvement. But for low-bias regions and episodes, it seems detrimental. Nevertheless, it is still a step in right direction: ignoring the emission reduction due to holidays to get smaller bias is just offsetting one error with another. One needs to disentangle the holiday-related error from the overall under-estimation, so that they can be handled separately. The current paper deals with the former issue.
- having the quantitative measure of improvement, we added and discussed a new Table 5 to the Results section showing the effect for all pollutants and major holidays.
- following the repeated criticism on the individual station's time series, we finally decided to remove those non-representative and/or hard-to-generalize examples.
- the style and language of the text have been reviewed and improved

**Specific comments**

**Reviewer 2**

Comment

An earlier comment about the names of the stations in figures 2-7 appears to has been slightly misunderstood. The suggestion was to include the station names (i.e. the station_name field in EEA terminology) rather than the station coordinates as identifier of location, which is not directly useful. Also, please include the types of the stations.

As regards figure 7 and the analysis thereof. This concerns a rural station (thus more likely to have a "background" $NO_2$ component rather than be directly affected by traffic) at the border with Germany. The authors mention in their response to an earlier comment that the corresponding discussion has been expanded to make the interpretation provided clearer, but these changes are not identifiable in the revised manuscript (version 4) in the specific section. In essence, based on the results presented, we have no direct way of positively attributing model (BL)-observation discrepancies seen between Dec 24-29 solely to reduced emissions. E.g. although this justifiably seems to be the case for Dec 25-26, the model-observation discrepancy for Dec 27-29 can only be explained with the additional hypothesis of emissions being influenced further to the formal public holidays. This may be the case to a certain extent (fig 11

seems to suggest such a phenomenon, albeit for a different country and mainly for 2019), but authors should clearly highlight the intricacies of drawing these conclusions based on results at a single station. In other words, it should be underlined that the episodic (i.e. meteorology driven) characteristics of this period complicates the interpretation of those time series, although hints on the impact of emissions are also possibly identifiable.

Response

The individual-stations examples have been criticized in several reviews, primarily for the lack of representativeness and generalization options (every case is indeed specific), so we finally decided to remove them all. The message of the paper is now concentrated on the region-, country-, and Europe-scales, as shown in the Table 5, Figures 2-5 for Europe and Figures 6-8 for regions and countries. The peculiarities of an individual station are of little interest unless they are supported by other stations in the region. We hope that it streamlined the presentation and made it more concise and to the point.

**Reviewer 3**

Comment:

The impact of the three simulations is shown only for temporal correlation coefficient of hourly $NO_2$ , CO, $O_3$ , and $PM_{2.5}$ concentrations (Figs.8-10) and for $NO_2$ concentration during Christmas period at one station: NL00107 (Fig.7). Figs. 2-6 show daily statistics for $NO_2$ concentration only for BL and HS simulations during Christmas, New Year, Easter, May and Ramadan holidays.

The purpose of this study to investigate at EU scale, in a systematic way, the effect of the holidays' emissions on air quality models' predictions is interesting, yet the manuscript does not show in comprehensive and concise manner that. The reader would like to see the impact of emissions on $NO_2$ , CO, $O_3$ , and $PM_{2.5}$ concentrations, at stations and over whole Europe, in the manuscript, not in the supplementary material where specific results at stations can be shown.

Response:

Thank you for the outline of the problems! The new Table 5 now presents the effect for all pollutants and all stations in Europe, in a harmonized quantitative way. The examples of the specific stations have been removed due to their low / unclear representativeness. The effect on individual stations is presented in maps of the Figures 6 and 7 and, in a country-aggregated form, Figure 8.

Comment

The study also does not show and discuss the relation between concentrations and emissions' reductions (HS and R3 scenarios) as a whole and by country as a marker for "regional specifics". Moreover, an evaluation of HS and R3 assumptions at stations, by country and station type can give indications about the value of the hypotheses used globally.

Response

There must be some confusion: the section "Regional specifics" and figures there discuss these very points. The section title has been revised to "Regional specifics of the effect of HS and R3 emission reduction" and changes were introduced into the text to highlight this relation.

Comment

In addition to a more careful analysis of the results, their presentation and discussion should be improved, both for language and rationale.

Response

The paper has been carefully read through correcting the language

Comment

Also not all claims are justified. For example "line 185 The impact of holidays on the SILAM spatial skills was the largest for the Christmas week (Figure 2a)". This comment should be supported by Fig.2 where all the holidays should be shown. As it is, this statement seems to be valid for all pollutants.

Response

The statement is indeed valid for all pollutants, with some reservations for PM. It is now supported by the new Table 5.

Comment

Moreover, an analysis and discussion of results as a function of pollutant type, supported by data and images should be included in the study. For example, intuitively, it is expected that the impact of reductions O3 during spring will be different from winter.

Response

The new Table 5 and related discussion now provide an overview of all pollutant types and highlight the ozone specificity.